# Unveiling and understanding health inequalities: A bi-clustering study on SDG3 implementation in the Italian regions

Monica Giancotti[1,2]*, Milena Lopreite[3,4], Marianna Mauro[2,5], Michelangelo Puliga[6]

1 Department of Law, Economics and Social Science, Magna Graecia University, Catanzaro, Italy,
2 Health and Innovation (H@I) Research Center, Department of Clinical and Experimental Medicine, Magna Graecia University, Catanzaro, Italy, 3 Department of Economics, Statistics and Finance, University of Calabria, Cosenza, Italy, 4 Institute of Management, Scuola Superiore Sant'Anna, Pisa, Italy, 5 Department of Clinical and Experimental Medicine, Magna Graecia University, Catanzaro, Italy, 6 Department of Economics, Ca' Foscari University of Venice, Venice, Italy

* mgiancotti@unicz.it

## Abstract

Health equity is a key policy priority in public health and a central component of Sustainable Development Goal 3 (SDG 3), which focuses on "Good Health and Well-Being". While SDG 3 sets global benchmarks, its local implementation—particularly in countries with decentralized healthcare systems and significant regional disparities—poses a considerable challenge. This study addresses the need for subnational analyses by moving beyond national averages and identifying region-specific barriers and enablers to achieving SDG 3 targets, using Italy as a case study. To this end, we apply spectral bi-clustering, an innovative data-mining technique, to regional SDG 3 indicators compiled by ISTAT for the years 2013–2019. The analysis pursues two objectives: (1) to identify clusters of Italian regions with similar SDG 3 profiles; and (2) to determine which indicators are most salient within each cluster and how they diverge from national benchmarks, deriving policy implications tailored to each group of regions. Our findings reveal three distinct regional clusters: the analysis demonstrates that certain health indicators are more relevant within specific regional contexts, pointing to structural and systemic variations in healthcare provision and outcomes. These results underscore the inadequacy of uniform policy approaches and highlight the need for regionally differentiated strategies. This study provides one of the first applications of spectral bi-clustering to health equity analysis at the subnational level, offering actionable insights for policymakers seeking to localize SDG 3 implementation and bridge health gaps across regions.

**Data availability statement:** All relevant data are within the paper and its Supporting Information files.

**Funding:** The author(s) received no specific funding for this work.

**Competing interests:** The authors have declared that no competing interests exist.

## 1. Introduction

Health equity is a primary policy objective in public health, aiming to ensure that all individuals have the opportunity to achieve good health by eliminating unfair and avoidable disparities in healthcare, social conditions, and physical environments [1–3]. In 2015, the urgent need to address these inequalities led 193 nations to commit to the United Nations Sustainable Development Goals (SDGs)(https://unstats.un.org/sdgs/indicators/Global-Indicator-Framework-after-2024-refinement-English.pdf), designed to guide global development through 2030 [4–7]. The SDGs are a crucial step beyond the previous Millennium Development Goals, as they explicitly address social disparities and operate on the principle that no one should be left behind [8]. The WHO European Health Equity Status Report underscores that the key conditions for a healthy life are directly aligned with several SDGs, including universal health services (SDG 3), basic income (SDGs 1 and 10), and decent living and working conditions (SDGs 2, 6, 7, 8, and 11) [9]. The call for equity is most prominent in the SDG3 goal (*Good Health and Well-Being*), which emphasizes universal health coverage and access to essential healthcare services as means to reduce health inequalities [10].

Despite these international insights, implementing the SDGs at a national and local level remains a complex challenge [11–14]. Evidence on SDG implementation points to difficulties in coordination and integration and highlights the absence of operational guidance for translating global goals into subnational action plans: goals are often viewed as too broad, focusing on national and global averages without accounting for significant within-country distributional inequalities [14–16]. Consequently, there is a lack of clear guidance for local governments on how to align global objectives with specific regional contexts [14].

Notably, the third goal (SDG3) centers on building a universal healthcare system that ensures healthy lives and wellbeing.

At the European level, some studies have assessed SDG 3 through cross-national dashboards and inequality monitoring frameworks [4,17]. Comparative analyses highlighted substantial heterogeneity across Member States, revealing clusters of strengths and persistent gaps often hidden by national averages [17]. Country-specific evidence reinforced these findings: Sweden's longstanding path toward universal health coverage illustrates how system design shapes SDG 3 delivery [18], while Spain-focused studies, drawing on GBD indicators and regional well-being metrics, show how resource allocation and risk factors translate into territorial disparities [19]. In essence, to implement SDG3 targets at national level, policies must be adapted to diverse regional contexts before healthy lives and well-being can be achieved and sustained across populations [14,20,21]. However, much of the empirical assessment of SDG 3 progress has concentrated on national scores or cross-country comparisons which—though informative—mask within-country heterogeneity crucial for designing appropriate policies [11,12,17]. Current SDG 3 monitoring efforts largely aggregate indicators to country means or produce dashboards that do not systematically identify "where" and "which" targets are underperforming within countries [4,10,17].

In practical terms, policymakers need evidence and methods to (i) identify groups of territories that share similar challenges in achieving SDG 3 targets, and (ii) isolate the indicators most responsible for underperformance within each group. Prior EU-wide and national case studies either (a) remain at the national scale, or (b) describe subnational disparities without pinpointing the subset of SDG 3 indicators that most strongly differentiate such groups [17,19]. This leaves a clear research gap: subnational, indicator-sensitive analyses are needed to go beyond averages and to reveal "which indicators matter, where" in a way that is directly actionable for regional planning.

This need is particularly acute in countries with decentralized health systems and marked regional differences [22–28]. Within this landscape, Italy represents a particularly instructive case. Although Italy's National Health Service is guided by the principles of universality and equity, the decentralization of healthcare administration, initiated in 1992, unintentionally created 21 distinct and uneven regional health systems [29,30], leading to substantial regional health disparities, especially between the more prosperous North and the South [31–40]. These disparities were exacerbated by financial deficits in some regions, leading the central government to impose strict, austerity-focused Recovery Regional Plans (RRPs) in 2007 [41,42]. While these plans stabilized regional finances, they did so at the cost of reduced healthcare services and resources [43,44]. Pre-existing disparities were further exposed and compounded by the COVID-19 pandemic. The pandemic highlighted how factors such as non-communicable diseases, poverty, and unemployment varied significantly across regions, deepening existing vulnerabilities [31,45].

Addressing these persistent health gaps requires reforms that deliver targeted interventions and equitable access to care tailored to local needs.

Recent Italian and WHO reports—documenting the persistence of territorial gaps in health outcomes—confirm the relevance of regional-scale analyses to support policy development and decision-making appropriate to diverse territorial contexts [31,46]. The recent ISTAT report likewise underscores the centrality of a subnational reading of the SDGs in Italy, particularly Goal 3 [47]. In line with the "no one left behind" principle, it reinforces the focus on territorial inequalities. Regional analysis shows a heterogeneous picture, with generally better performance in the North-East and delays in the South and Islands; despite signs of recent improvement in some southern regions, progress still does not bridge the historical gaps with the North. These findings strengthen the case, in the Italian context, for regional analyses and targeted policies to fully translate SDG 3 into equitable and context-sensitive health outcomes [47].

In addition, recent Italian evidence has reinforced the link between the "design" of health services and health outcomes, while highlighting pronounced local variation [48,49]. Studies also point to a non-negligible organizational–geographic component, suggesting that differences in the design and management of regional services (e.g., supply, waiting times) translate into disparities in utilization and, ultimately, outcomes. Finally, evidence on fiscal and organizational decentralization suggests a risk of amplifying pre-existing inequalities in access to, availability and use of services, and in health outcomes, when resources are devolved without a full understanding of regional characteristics, needs, and gaps [50].

This recent evidence—centered on design–outcome relationships, financial barriers, and local variation—justifies our subnational approach: it is essential that research examine differences among regional health systems to understand how global objectives such as SDG 3 can be effectively translated into meaningful social change. By monitoring health outcomes at the regional level, policymakers can identify obstacles to SDG implementation and promote effective, equity-oriented policies. In this sense, there is a gap in the literature that this study seeks to fill by offering a subnational perspective that responds to calls to move beyond averages and to identify context-specific barriers and enablers for the implementation of SDG 3 in Italy.

Specifically, we apply an innovative data-mining technique—bi-clustering [51]—to ISTAT's regional SDG 3 indicators (2013–2019) in order to: (1) identify groups of Italian regions with similar SDG 3 profiles; (2) determine, for each group, which indicators are most salient (i.e., most strongly expressed) and how they deviate from national averages; and (3) translate these findings into cluster-tailored implications for health policy. These aims are articulated in two research questions (RQs):

1. *How can bi-clustering of SDG 3 indicators reveal distinct groups of Italian regions, and what insights do these clusters provide about regional health-related trends?* The innovation of the bi-clustering algorithm lies in its ability to investigate the nuanced interaction between regions and SDG3 targets. Unlike traditional clustering, which groups regions based on all variables, bi-clustering allows us to examine the "partial expression" of a variable within a specific cluster. This means we can explore the relevance of a particular SDG3 target across different groups of regions, even when that variable does not represent a strong or weak signal in the overall dataset. This capability is crucial for policymakers because it helps pinpoint region-specific factors shaping health outcomes that might otherwise be obscured by national or broad regional averages.

2. *Which SDG 3 variables within each cluster deviate most from the national average, and what policy implications follow from these disparities?* Addressing this second question enables us to draw direct conclusions about potential barriers to achieving SDG3 within each group of regions (*clusters*). This is fundamental to the SDG agenda, which emphasizes that no one should be left behind. By identifying these critical deviations, our research provides the evidentiary basis needed to design health policies or strategies that account for the specific constraints of each territory. This enables policymakers to craft genuinely "tailored" interventions aimed at achieving SDG3 targets, moving beyond a standardized approach and promoting genuine health equity.

The paper is structured as follows: the next section describes the method. Empirical results are presented in section 3. We then discuss the findings (section 4) and draw general conclusions, including policy recommendations (section 5).

## 2. Materials and methods

### 2.1. Bi-clustering technique

Data mining algorithms are useful for processing large data sets to extract relevant information and knowledge [52]. Bi-clustering is a data mining technique created to study variables that do not (fully) express across different sample groups. The aim is to identify clusters of data grouped according to their similarities.

In 1972, the methodology was introduced by Hartigan [53] and later was used first by Dhillon [54] in the context of pure computational research, and then by Kluger et al. [55] in genomics. In the field of bioinformatics, this method is now well developed, and a recent survey of Castanho et al. [51] shows all the advancements in this discipline. However, in the economic literature there are still few works using bi-clustering: notably we find the works of Chao et al. [56] and Haedo [57], that used bi-clustering to investigate the interplay between regions and industrial sectors by allowing to examine the "partial expression" of a specific industrial sector in different groups of regions. The main advantage of bi-clustering compared to traditional clustering is that the latter operates in one dimension, i.e., grouping objects in clusters homogenous according to a single variable or a combination of variables expressed as a vector. Bi-clustering, instead, is bi-dimensional: objects are grouped simultaneously using two dimensions. The easiest way to visualize and understand bi-clustering is through matrices, where bi-clusters appear in a checkerboard structure with rows and columns having similar properties. A bi-dimensional structure reveals specific patterns traditional clustering often misses, identifying variables particularly relevant in individual groups.

Compared to panel methods, bi-clustering helps discover patterns where time is not relevant, or when the time span is insufficient for panel analysis. Another advantage is that no assumptions are made on linearity, independence, or inner correlation of data as in panel methods—there's no need to test for collinearity between variables that may bias estimation.

From this perspective, bi-clustering is not just a novel application; it is a promising methodological tool that generates more robust and statistically reliable findings because it overcomes limitations inherent in both traditional econometric methods (like panel analysis) and standard unsupervised clustering (like k-Means), making it a promising tool for

regional-level research. Specifically, compared to standard panel data methods, bi-clustering offers major robustness by avoiding restrictive assumptions relative to:

- Assumption-Free Analysis: Bi-clustering requires no assumptions regarding linearity, independence, or inner correlation of the data. This eliminates the necessity for complex diagnostic testing (such as those for collinearity or endogeneity) that often complicates and biases estimations in econometric panel models.

- Time-Insensitivity: It is highly effective in contexts where the time dimension is insufficient for reliable panel analysis, or where the research focus is primarily on spatial heterogeneity rather than temporal dynamics.

Moreover, bi-clustering significantly enhances pattern discovery by leveraging the concept of "partial expression," which is crucial when variables are numerous, and observations are relatively scarce (both spatially and temporally) as follows:

- Bidimensionality for Less Noise: Traditional methods (like k-Means) create unidimensional clusters (grouping regions based on all indicators), forcing relationships to hold across the full dataset. In contrast, Spectral Bi-clustering [55] simultaneously clusters rows (Italian regions) and columns (SDG3 indicators). This bi-dimensionality ensures that bi-clusters only include SDG3 indicators that are more similar and constant (low variance) within that specific subset of Italian regions. This focused approach systematically reduces noise and maximizes pattern coherence.

- Capturing Localized Phenomena: This ability to discover "partial expressions" is key for regional research. A variable (e.g., an SDG3 health indicator) may be highly relevant in a subset of Italian regions, but not across the entire Italy. Bi-clustering inherently captures these localized relationships (e.g., specific regional dependencies, like border regions benefiting from foreign health facilities) that would otherwise be masked by national averages or require the manual addition of complex dummy variables in traditional models.

Starting from this point and in the context of the SDG3 goals, we applied bi-clustering methods using a bi-dimensional dataset where SDG3 indicators are the columns and regions are the rows. Our goal is to find specific groups of SDG3 variables more relevant for different groups of Italian regions. The bi-clustering algorithm extracts useful information, including: (a) understanding SDG3 performance across clusters of Italian regions; (b) providing useful feedback to policymakers about regional characteristics linked to similar SDG3 performance; and (c) identifying potential barriers to attaining SDG3 goals, with consequences for policy effectiveness across regions. Specifically, we began the analysis by making an average of the SDG3 indicators from the ISTAT database over the 2013–2019 period. For each Italian region, we computed a single value for every SDG3 variable by averaging the indicator values over these years (see Supporting Information - S1 Table). Using this dataset, we applied a bi-clustering algorithm to identify clusters of Italian regions. The algorithm requires a pre-specified number of clusters to be formed. Starting from the work of Gandullia and Leporatti [58], we considered three homogeneous groups, which align with the structural and organizational differences of the Italian Regional Health Systems (RHSs) and their socio-economic characteristics [58]. This distinction is based on factors such as per capita healthcare expenditure, managerial efficiency, and the quality of services, suggesting a natural segmentation into northern, central, and southern macro-areas. Accordingly, we imposed the number of clusters K = 3 to identify which SDG3 variables are most influential within each regional cluster. We applied the Spectral Bi-clustering algorithm [55] on the data from S1 Table to achieve this.

To maintain the integrity of our unsupervised approach, data pre-processing was kept to a minimum. We used data from 2013 to 2023 but excluding COVID-19 years (2020–2023). We treated this seven-year period as "homogeneous" by calculating the average value per indicator for each region. This approach ensured that our final dataset contained no missing values and retained the original unit of measure for each variable.

Before clustering, a z-score transformation (centering the mean and scaling by the standard deviation) was applied to the data. This crucial step ensured that all variables were on the same scale, which is a prerequisite for reliable clustering

and allows the algorithm to form meaningful bi-clusters of both indicators and regions. We compared the performance of bi-clustering against a traditional method, K-Means, using two key tests. We used K-Means and we applied the same number of clusters to our bi-clustering model (Supplementary Materials S2 Table).

The first test evaluated each method's ability to identify meaningful groups of regions based on their socio-economic characteristics and statistical power. We used the ANOVA f-test to assess for significant differences between cluster means (S4 and S5 Tables in the Supporting Information show F-value and p-value). In addition, we performed, for the same goal, the Kruskall-Wallis test to provide an alternative comparison (S2 and S3 Tables in the Supporting Materials):

K-Means divides regions based on all health indicators, creating groups with similar means and standard deviations. In fact, the algorithm works by minimizing the within-cluster variance (inertia) by finding the centroids of data points using Euclidean distances [59].

Bi-clustering, instead, by allowing for partial expression of variables, is able to separate not only Italian regions (rows), but also SDG3 health indicators (columns) making it feasible to create more independent clusters where independence can be seen (at least partially) with ANOVA tests and the Tukey-HSD group test.

In summary, statistical tests show that Bi-clustering builds better separation, and (partial) cluster independence (see for more details S4- S7 Tables in the Supporting Information, where cluster 0 is different from cluster 1 and 2, but cluster 1 is not independent from cluster 2, with ANOVA and Tukey-HSD). Conversely, K-Means tends to rearrange members in order to have a smaller within – cluster variance (inertia) and a general smaller in-cluster variance renouncing to have independent clusters. While statistical tests provide valuable insights, we believe a more complete understanding comes from evaluating the results against real-world socio-economic conditions in Italy.

The traditional division of Italy into North, Center, and South, combined with knowledge of regional healthcare systems, strongly supports the partitions created by bi-clustering.

For example, bi-clustering grouped Valle d'Aosta, Trento, and Bolzano/Bozen together, which is a more accurate reflection of their similar socio-economic history than the K-Means result, which only clustered Bolzano/Bozen.

In conclusion, our comparison with K-Means demonstrates two significant advantages of using bi-clustering for this dataset: a) it produces clusters with greater statistical separation and partial independence due to its ability to use only a subset of relevant variables.; b) it accurately reflects the socio-economic geography of Italy's regions, producing more meaningful and interpretable clusters. The analysis was performed using the Python Scikit-learn library, specifically the Spectral Bi-clustering using the default settings, with the "scale" method for row/column normalization.

This setting applies two different normalizations one over rows and another over columns, a choice justified by assuming that the variability internal to the indicators could be different of the variability along the regions.

Moreover, the K-Means methods have the only parameter n_clusters = 3. We remember that before applying any clustering or bi-clustering algorithm data of each indicator have been transformed with z-score to have comparable scale for each variable. This to avoid the pre-dominance of some indicator given its unit of measure.

Finally, we investigated the "size effect" of each cluster by computing the Cohen's d effect size measure for bi-clustering and k-means. Based on this metric, the bi-clustering results yield stronger distinctions: all bi-clusters demonstrated at least a medium effect size, with clusters 0 and 2 achieving a larger effect size. Conversely, all k-Means clusters show a small effect size, indicating that their group averages are too similar to indicate a meaningful difference (S8 Table in the Supporting Materials).

## 2.2. Data

The selection of SDG3 indicators was guided by both conceptual relevance and data availability in the Italian context.

First, we included all SDG3 indicators officially reported by ISTAT [60], which provides harmonized regional-level data consistent with the United Nations framework.

This ensured that the analysis was grounded in indicators already validated for monitoring progress toward SDG3 at the national level.

Second, we excluded indicators that are not applicable to Italy, such as those related to tropical diseases (e.g., Malaria and Tuberculosis), since they are not measured or relevant in the Italian epidemiological context. This exclusion avoided introducing variables with no variance across regions, which would have biased the clustering results.

Third, we focused on indicators available for a sufficiently long- and homogeneous-time span (2013–2019), excluding years affected by the COVID-19 pandemic (2020–2023) to prevent distortions caused by this exceptional shock. The list of indicators for Italy is detailed in Table 1, and with full database in Supplementary Material (S1 Table), respect to the general international version, there are no indicators on tropical diseases (Malaria and Tuberculosis).

**Table 1. Regional Health Disparities in Italy: Bi-Clustering Analysis of SDG3 Indicators Compared to National Averages.**

| Indicator | Unit of Measure | bi_clusters | ratio_bc_country |
|---|---|---|---|
| Age-specific fertility rates per 1000 women aged 15–19 | Births per 1,000 women | 0 | 0.900787 |
| Alcohol (standardized rates) | Per 100,000 population | 0 | 1.030595 |
| Standardized suicide mortality rate | Deaths per 100,000 population | 0 | 1.005708 |
| Standardized mortality rate for accidental poisoning | Deaths per 100,000 population | 0 | 1.032183 |
| Probability of dying between the ages of 30 and 69 from cancer, diabetes, cardiovascular and respiratory diseases | Percentage | 0 | 0.993060 |
| Pharmacists | Per 10,000 population | 0 | 1.063578 |
| Percentage of births with more than 4 check-ups performed during pregnancy | Percentage | 0 | 1.064548 |
| Doctors | Per 10,000 population | 0 | 1.027815 |
| Age-specific fertility rates per 1,000 women aged 10–14 | Births per 1,000 women | 1 | 2.461635 |
| Smoking (standardized rates) | Per 100,000 population | 1 | 1.005910 |
| Road accident fatality rate | Deaths per 100 road accidents | 1 | 0.915553 |
| Rate of serious injury in road accidents | Injuries per 100 road accidents | 1 | 0.816293 |
| Nurses and midwives | Per 10,000 population | 1 | 0.849746 |
| Number of deaths in road accidents | Number of deaths | 1 | 1.349929 |
| Influenza Vaccination Coverage Age 65+ | Percentage | 1 | 1.055987 |
| Incidence of HIV infections per 100,000 residents (by region of residence) | Cases per 100,000 population | 1 | 0.757329 |
| Healthy life expectancy at birth | Years | 1 | 0.957185 |
| Excess weight (standardized rates) | Per 100,000 population | 1 | 1.124619 |
| Diabetes (standardized rates) | Per 100,000 population | 1 | 1.264639 |
| Dentists | Per 10,000 population | 1 | 0.856771 |
| Beds in ordinary hospitalization in public and private healthcare institutions | Beds per 100,000 population | 1 | 0.885674 |
| Arterial hypertension (standardized rates) | Per 100,000 population | 1 | 1.161467 |
| Neonatal mortality rate | Deaths per 1,000 live births | 2 | 1.003174 |
| Probability of death under 5 years | Per 1,000 live births | 2 | 0.942874 |
| Day-Hospital beds in public and private healthcare institutions | Beds per 100,000 population | 2 | 0.922460 |
| Contraception demand satisfied with modern methods during the last 12 months | Percentage | 2 | 1.128400 |
| Childhood vaccination coverage: polio | Percentage | 2 | 0.993593 |
| Beds in residential socio-assistance and socio-health facilities | Beds per 100,000 population | 2 | 1.590647 |
| Vaccination coverage in childhood: measles | Percentage | 2 | 0.983885 |
| Vaccination coverage in childhood: rubella | Percentage | 2 | 0.984377 |

**Note:** Three different bi-clusters grouping Italian regions and SDG3 variables (the average value of each SDG3 variable in the cluster is compared to the national average).

Pre-processing of data has been limited to the minimum in the spirit of using the Unsupervised method. As data refers to the years 2013–2019 (included) we consider these 7 years as "homogeneous" by taking the averages (over 7 years) per indicator, per region. This in turn will produce a table that has no missing data and it retains the same unit of measure of the initial variables.

Clustering, however, cannot reliably work with variables that are expressed in different scales: a simple z-score transformation (removing mean and dividing per the standard deviation) has been applied before clustering.

This way all variables have the same scale and clustering will select indicators and regions to form bi-cluster combinations.

## 3. Results

### 3.1. Clustering model

The bi-clustering analysis successfully generated three distinct clusters of Italian regions, labeled as Cluster 0, Cluster 1, and Cluster 2. Each one of these clusters is associated with a specific subset of relevant SDG3 indicators that have the most influence on the regions within that group.

This allowed us to answer our first RQ (RQ1): "*How can bi-clustering of SDG 3 indicators reveal distinct groups of Italian regions, and what insights do these clusters provide about regional health-related trends?*". The groups of regions discovered through this blind and unsupervised method are shown in Fig 1.

### 3.2. Analysis of SDG3 trends across identified cluster

By applying a bi-clustering approach to SDG3 indicators, our analysis highlights regional disparities and identifies context-specific health system barriers in the Italian regions. This approach allows us to move beyond national averages

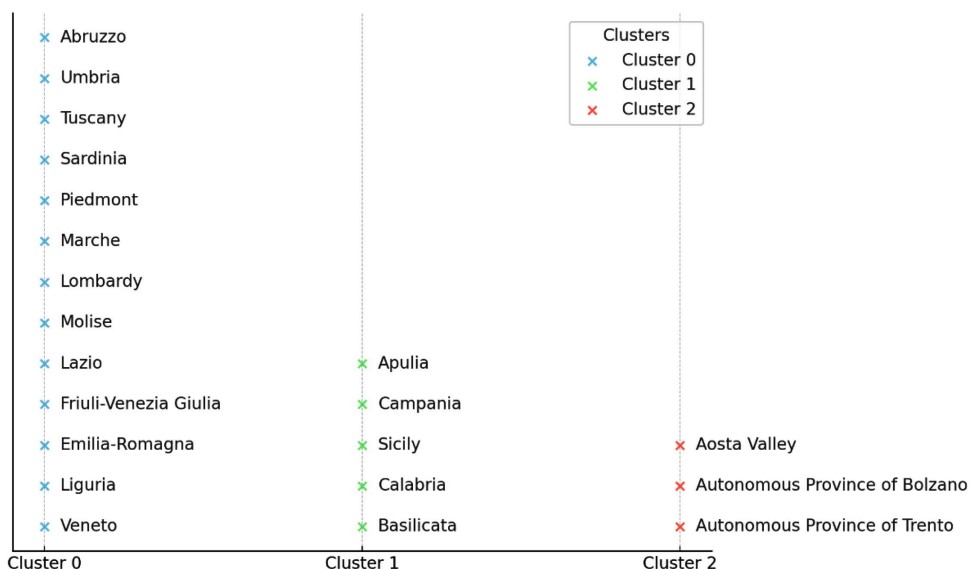

**Source:** *Authors' elaboration.*

**Fig 1. Italian Regions grouped using bi-clustering approach.**

and identify distinct regional challenges and strengths. As shown in Fig 1, our study divides Italian regions into three distinct clusters based on their health-related trends, as detailed in Table 1.

The table shows three main columns as follows:

- *Indicator*: It shows the specific health-related variable or SDG3 indicator;

- *bi_clusters*: It reports the cluster of the variables/regions;

- *ratio_bc_country*: It is the ratio of the average value of a specific SDG3 variable in each cluster compared to the national average.

In detail, a value greater than 1 suggests that the cluster average value for this indicator is above the national average, while a value below 1 indicates that it is below the national average. The composition of each cluster is determined by the algorithm, grouping regions that exhibit similar behavior across SDG3 variables.

By comparing the average value of each variable within a cluster to the national average (represented as 1 or 100% by default), we can identify specific regional disparities. Specifically, by investigating the prevalence of one or more SDG3 variables explaining each cluster we computed the average value of each SDG3 variable in a specific cluster and then we divide it by the national average of this variable (RQ2: "*Which SDG 3 variables within each cluster deviate most from the national average, and what policy implications follow from these disparities?*"). For instance, in Cluster 0, the variable "Pharmacists" shows a value of 1.064, which means the average number of pharmacists in the regions belonging to this cluster is 6,4% higher than the national average. Conversely, a value of 0.849 for "Nurses and midwives" in Cluster 1 implies that the average number of nurses and midwives in these regions is 15% lower than the national average.

Examining the trends of the SDG3 variables across the three clusters reveal the following interesting insights:

- Cluster 0 shows a broad variety of SDG3 variables (in the sense that they represent diverse, and unrelated aspects of the health conditions of the Italian population like accidental poisoning, and fertility rates) with several values that remains close to the national average (=1 or 100% by default). This cluster, however, has an above average number of doctors (+2%) and pharmacists (+6%), which likely contribute to its relatively balanced health indicators.

- Cluster 1 – composed predominantly of Southern Italian regions – faces a critical strategic gap. This cluster shows a high incidence of diabetes (26% more than the national average), together with excess weight (12% more than the national average), a lower life expectancy (5% lower than the national average) combined with a concerning decline in healthcare resources, including 12% fewer beds in ordinary hospitals and fewer nurses compared to the national average.

These findings point to a weaker healthcare system, likely affected by past spending cuts and a poor level of preventive care.

On the contrary, other indicators, such as fertility rates, show instead an above average value suggesting a potential minor future pressure on the health system for population aging.

- Cluster 2 is formed by the two Autonomous provinces of Trento and Bolzano, and Valle d'Aosta stands out for its robust healthcare infrastructure.

It has a significantly higher number of beds in private care facilities (+59%) and a greater reliance on traditional full-length hospitalization (+59%) over day-hospital services (−8%).

This suggests a strategic choice to invest in comprehensive, long-term care. This cluster also shows a lower probability of death for children under five (−5%), indicating likely better pediatric assistance.

Our findings provide a clear roadmap for SDG3 implementation in Italy by moving from data-driven insights to actionable policy recommendations:

a) *Targeted Investment*: Cluster 1 highlights the urgent need for a strategic shift toward increased funding for prevention programs to combat diseases like diabetes. It shows that current financial RRPs have been ineffective at improving health equity and that these regions require tailored financial support to align with SDG3 goals;

b) *Focus on Human and Physical Resources*: the above-average number of doctors and nurses in Cluster 0 underscores that human capital is a significant driver of positive healthcare quality. Similarly, the high number of beds in Cluster 2 highlights the strategic importance of investing in a balanced care network, including both high- and low-complexity facilities, to ensure sustainability and equitable access, especially for an aging population. In summary, our bi-clustering approach identifies key regional priorities that traditional methods may underestimate.

It shows that effective, equitable healthcare reform in Italy requires a multi-pronged strategy that is specifically tailored to the unique needs of each region.

### 3.2.1. Intra-cluster similarities and inter-cluster divergences: The role of infrastructure variables and cost containment policies.

The last phase of the analysis started from the three clusters in order to identify the factors that explain most of the similarities/divergences in the variability of the SDG3 targets among Italian regions.

The bi-clustering application has a key advantage: it uncovers patterns not visible with other techniques, selecting SDG3 variables based on their prominence in each regional group.

This allows us to see that while all SDG3 indicators exist in every Italian region, only some are truly important in each cluster.

As shown in Fig 1, our analysis divides Italian regions into three distinct clusters based on their health-related trends, with the results summarized in the checkerboard plot of Fig 2.

This plot visually compares each cluster's performance against the national average, making regional differences immediately apparent.

Specifically, in Fig 2, Table 1 is depicted by the *checkerboard plot* that shows each cluster through a two-color scheme, using blue to indicate the value of the indicator below national average and red for a value above the national average, allowing us to immediately capture the differences between the groups.

The three identified clusters display intra-cluster (within the same group) similarities and inter-cluster (between different groups) divergences, both in terms of levels of achievement of SDG3 objectives and in terms of infrastructural characteristics of the RHSs and cost containment policies, suggesting a potential relationship between these variables.

In particular, Cluster 0 consists of regions with a variable distribution of SDG3 indicators that are on average, very close to the national standard. This cluster's relative stability appears to be supported by a higher-than-average number of doctors (+2%) and pharmacists (+6%). Regarding cost containment policies, Cluster 0 includes regions from the North, the Center, and Sardinia.

The cluster has a varied composition as most of the regions included in this cluster (9 on 13) have not experimented with RRPs in the analyzed period; 3 regions (Abruzzo, Molise and Lazio) were under the effect of RRPs; finally, Piedmont started the RRP in 2013 and finished in 2015.

In general, the inclusion of regions both with and without RRPs suggests that their healthcare systems, benefiting from greater infrastructural stability, were more resilient to the effects of cost-containment.

Cluster 1 includes the regions of Southern Italy.

These regions face significant challenges in achieving SDG3 objectives, specifically they show: a higher incidence of diabetes (+26%), excess weight (+12%), and a lower life expectancy (−5%). Cost containment policies seem to have exacerbate the health gap compared to other Italian regions. Indeed, Cluster 1 includes those regions that are or have been under the RRP programs for an extended period, with the exception of Basilicata Region (https://www.salute.gov.it/portale/pianiRientro).

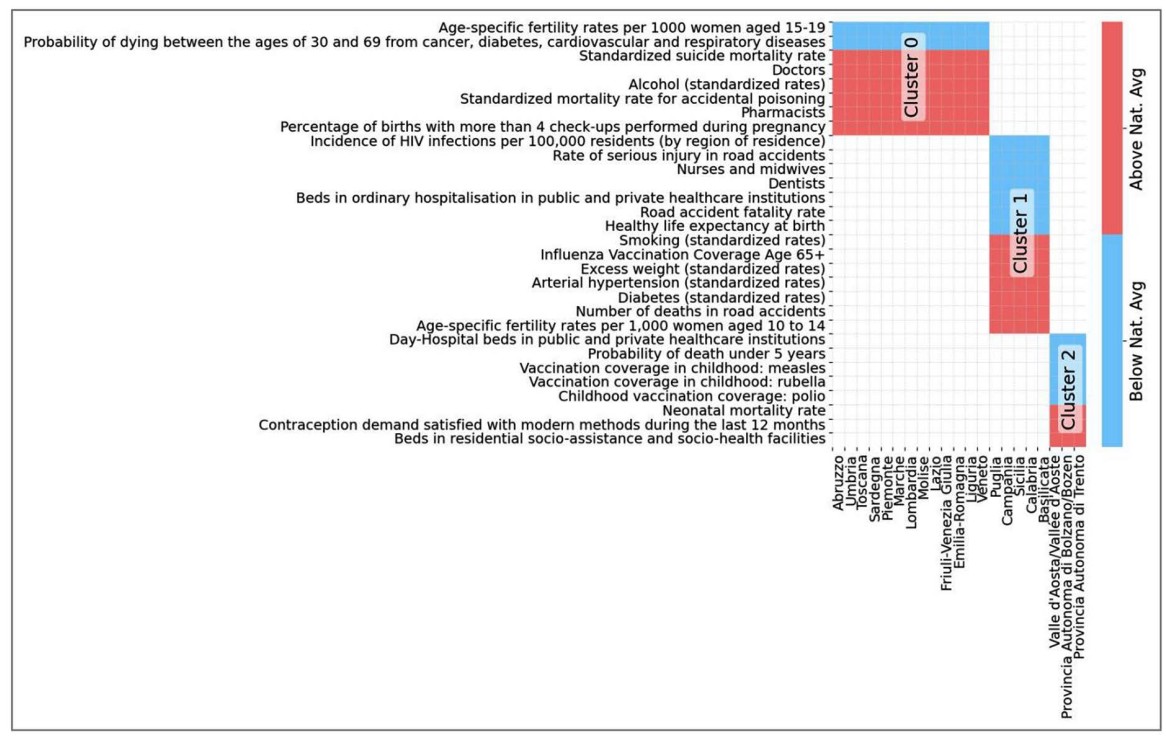

**Source:** Authors' elaboration. **Note:** The bi-clusters rows/columns indicate the association between Italian regions (*Cluster 0:* Veneto, Liguria, Emilia-Romagna, Friuli-Venezia Giulia, Lazio, Molise, Lombardy, Marche, Piedmont, Sardinia, Tuscany, Umbria, Abruzzo. *Cluster 1:* Basilicata, Calabria, Sicily, Campania, Apulia. *Cluster 2:* Autonomous Province of Trento, Autonomous Province of Bolzano, Valle d'Aosta) and the prevailing SDG3 indicators in each cluster. The colors show that the SDG3 indicator in a cluster is below (blue) / above (red) national average.

**Fig 2. Checkerboard representation of the matrix "SDG3 goals versus Italian regions".**

The prevalence of diseases such as diabetes and arterial hypertension in these regions—together with excess weight is likely a result of inadequate policies for the primary care system.

Variables relative to the infrastructure of health systems also appear to have been negatively affected by cost containment policies. For example, the regions in Cluster 1 are characterized by evident shortcomings, with hospitals often undersized relative to the population, and a number of beds and healthcare personnel (nurses and midwives) below the national average (−12% and −15%, respectively).

Finally, Cluster 2, which includes the autonomous provinces of Trento, Bolzano and Valle d'Aosta, offers a clear example of effective strategic investment. These regions stand out for their robust healthcare infrastructure, particularly a high number of hospital beds in private and social care facilities (+59%). Their enhanced autonomy and ability to manage funds in a customized way have had a positive impact on health system performance, as evidenced by a 5% lower probability of under-five mortality compared to the national average.

Our findings provide interesting recommendations for SDG3 implementation in Italy by moving from data-driven insights to actionable policy recommendations:

a)  *Tailored Policy Design*: instead of a single national strategy, our results show that effective SDG3 implementation requires policies tailored to each region's specific needs;

b)  Specific and focused Investment: cluster 1 highlights the urgent need for a strategic increasing of funding for prevention programs and human capital to combat diseases and address infrastructure shortfalls;

c)  *Leverage Autonomy for Progress*: the success of Cluster 2 demonstrates that granting regions greater decision-making and financial autonomy can lead to strategic health investments that significantly improve SDG3 outcomes.

In summary, our bi-clustering approach identifying key regional priorities shows that effective, equitable healthcare reform in Italy must be a multi-pronged strategy that is specifically tailored to the unique economic, social, and cultural context of each region.

## 4.  Discussion

Our work responds to the critical need for research that analyses differences among regional healthcare systems to better understand how global objectives, such as SDG 3, can be effectively translated into meaningful societal change. By applying a bi-clustering approach to SDG3 indicators, our analysis highlights significant regional disparities in Italy's healthcare system and identifies context-specific health system barriers.

In particular, three distinct clusters emerge:

1)  Cluster 1 (Fig 2), which includes Southern Italian regions, faces a critical strategic gap. Our analysis reveals a concerning decline in healthcare resources, including non-day-hospital beds, doctors, and nurses, which exacerbates long-standing issues such as chronic poverty and inadequate infrastructure. The persistence of preventable diseases in this cluster highlights the urgent need for a strategic shift towards increased funding for prevention programs. Our findings suggest that, as things stand, current financial RRPs have been ineffective in improving health equity and that these regions require tailored financial support to align with SDG3 goals [29]. In this sense, concrete and measurable recommendations for regions in this low-performing cluster include: prioritizing (i) reduction of SDG 3.4 risks by lowering standardized diabetes rates from 1.26× to ≤ 1.00 × the national average and excess body weight from 1.12× to ≤ 1.00× within 5 years through funded primary care and lifestyle programs, (ii) strengthening the SDG 3.c workforce by increasing the number of nurses/midwives from 0.85× to ≥ 1.00 × the national average (+15%) within 3 years, (iii) restoring ordinary inpatient bed capacity from 0.89× to ≥ 1.00× per 100,000 population within 5–7 years, and (iv) achieving SDG 3.6 by reducing road-traffic fatalities from 1.35× to ≤ 1.00× within 3 years through law enforcement, emergency response networks, and remediation of crash hotspots.

    For regions in this cluster, progress should be verified annually using the *ratio_bc_country* metric reported in Table 1.

2)  Cluster 0 (Central, Northern, and Sardinian regions) provides a valuable model for achieving balanced SDG3 outcomes. The above-average presence of doctors and nurses in this cluster appears to be a key driver of positive healthcare quality, underscoring that human capital is a significant strategic factor in effective healthcare systems.

3)  Finally, Cluster 2 (Valle d'Aosta and Trentino-Alto Adige) offers a vital strategic lesson on the importance of infrastructure. The high number of ordinary hospital beds in this cluster suggests that adequate long-term care capacity is essential and cannot be fully replaced by day-hospital solutions, particularly given Italy's aging population. The discrepancies observed between Cluster 2 and Cluster 1 highlight the need for a balanced network of high- and low-complexity healthcare facilities, a strategic tool for ensuring both sustainability and equitable access to care [61].

 

At the national level, to promote convergence across all regions and reduce divergences among the three clusters, we recommend developing structured knowledge-transfer mechanisms from higher-performing clusters to lagging regions. To this end, it is essential to: (a) design action plans targeted to each region's priority SDG 3 gaps, (b) implement short, focused placements for clinical, managerial, and public-health staff to transfer routines and governance practices from better-performing regions to those lagging on one or more SDG 3 targets, (c) develop concise, adaptable protocols (prevention, workforce, and service delivery) aligned with local capacities, and (d) conduct recurring reviews of regional cases and learning sessions centered on the indicators showing the largest gaps. Together, these mechanisms convert observed strengths into reproducible routines while maintaining general, transferable, and measurable recommendations against SDG 3 progress.

In summary, our findings provide a clear and useful roadmap for SDG3 implementation in Italy. The bi-clustering approach helps identify key regional priorities that traditional methods might overlook, demonstrating how an innovative methodology can transform a global framework into concrete, localized policy interventions. In light of this evidence, international organizations can also use clustering to ensure that scientific evidence informs appropriate resource allocation. For example, countries could be grouped by co-occurring needs and delivery constraints, and financial resources could then be distributed so that clusters furthest from target outcomes receive higher per-capita allocations. Finally, the composition of clusters should be monitored annually, with entities reclassified as systems improve.

## 5. Conclusion

This study makes several key contributions by demonstrating the utility of bi-clustering as a powerful tool for analysing health data in a localized context. While traditional methods like standard clustering or panel data analysis can provide a broad overview, they often fail to capture the specific structural disparities present in heterogeneous systems like Italy's. Our approach addresses this by identifying "partial expressions" of complex variables, offering a more granular understanding of SDG implementation at the regional level. Our findings reveal how a decentralized health system, coupled with austerity measures and socio-economic disparities, has led to divergent health outcomes across Italy. We show that a one-size-fits-all approach is ineffective and that targeted governance measures are essential for each region.

Our study allows us to offer recommendations for different stakeholders.

First, in light of our results, national policy makers should channel greater investments toward initiatives that advance SDG 3, especially in underserved and economically disadvantaged regions, optimizing resource allocation to improve health-service management where needs are most urgent. We also underscore the adverse effects of the RRPs, which, while reducing regional debt, have inadvertently worsened health indicators.

The methodology we applied in Italy can serve as a valuable model for monitoring SDGs in other national settings, particularly those with significant internal disparities. For example, countries with decentralized or devolved governance structures—such as India, Brazil, or the United States—could use this method to identify which SDG-specific variables are most critical for their subnational regions. In doing so, policy makers could move beyond national averages and design interventions tailored to local needs.

For researchers, the bi-clustering approach is not limited to SDG 3; it can be extended to other indicators to reveal important regional differences in areas such as poverty (SDG 1), education (SDG 4), or access to safe drinking water (SDG 6), thereby supporting appropriate resource allocation and the development of more effective, targeted policies. In practical terms, it will be necessary to: (a) identify the clusters, (b) determine which diverge most from SDG targets and therefore require targeted action, and (c) repeat the analysis over time to observe how clusters change before and after reforms, and whether gaps narrow or widen.

Finally, we conclude with the following recommendations for international agencies: (a) link funding to subnational clusters rather than national averages, so that resources follow the largest gaps, (b) ask countries to publish simple quarterly dashboards that show each indicator relative to the national average, making progress visible and comparable, and (c)

create a learning loop by sharing models, code, and examples, enabling countries to replicate what works and correct what does not at the cluster level.

The study has limitations. First, the data for SDG3 targets in Italy only begins in 2013, and the available time series data is short (up to 2023). The COVID-19 pandemic (2020−2023) also severely disrupted previous trends, which forced us to restrict our analysis to the 2013−2019. While this choice was necessary to avoid distortions introduced by the extraordinary impact of the pandemic on health systems and SDG3 indicators (e.g., sudden reductions in hospital capacity, shifts in healthcare priorities, and changes in population health behaviours), it also implies that our results primarily reflect pre-pandemic dynamics. Consequently, the regional disparities identified in this study should be interpreted as representative of structural and long-term features of the Italian healthcare system, rather than of its performance under emergency conditions. Future research will be needed to assess how the pandemic has reshaped these disparities and whether the clusters we identified have persisted, converged, or diverged in the aftermath of COVID-19. Second, our use of spectral bi-clustering has limitations. This method is sensitive to data characteristics and can be unstable. As a "global" method, it may not be effective at identifying clusters that exist at different scales within the same dataset. Furthermore, our application, which evolves time series data, does not inherently account for the sequential nature of time, as it treats all time points (columns) as independent features. Finally, our conclusions are specific to the Italian context with its unique decentralized healthcare system and pronounced regional disparities. Therefore, these findings may not be directly transferable to countries with centralized healthcare systems or less significant regional differences.

## Supporting information

**S1 Table. Averaging SDG3 values for each region.**
(DOCX)

**S2 Table. Comparison of K-Means vs Bi-clustering.**
(DOCX)

**S3 Table. Kruskall-Wallis test on cluster independence.**
(DOCX)

**S4 Table. ANOVA test on cluster independence for bi-clustering groups.**
(DOCX)

**S5 Table. ANOVA test on cluster independence for k-means groups.**
(DOCX)

**S6 Table. Tukey-HSD test for independent groups for bi-clustering.**
(DOCX)

**S7 Table. Tukey-HSD test for independent groups for k-Means.**
(DOCX)

**S8 Table. Cohen's d effect size for bi-clustering and k-Means.**
(DOCX)

**S1 File. Data.**
(XLSX)

**S2 File. Biclustering results.**
(CSV)

## Author contributions

**Conceptualization:** Marianna Mauro.

**Data curation:** Marianna Mauro, Milena Lopreite, Michelangelo Puliga, Monica Giancotti.

**Formal analysis:** Milena Lopreite, Michelangelo Puliga.

**Methodology:** Milena Lopreite, Michelangelo Puliga.

**Supervision:** Monica Giancotti.

**Visualization:** Monica Giancotti.

**Writing – original draft:** Monica Giancotti, Milena Lopreite, Michelangelo Puliga, Marianna Mauro.

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
