## [Decision Letter · Decision Letter 0]

4 Aug 2025

PONE-D-25-29410Unveiling and Understanding Health Inequalities: A Bi-Clustering Study on SDG3 Implementation in the Italian regionsPLOS ONE

Dear Dr. Giancotti,

Thank you for submitting your manuscript to PLOS ONE. After careful consideration, we feel that it has merit but does not fully meet PLOS ONE’s publication criteria as it currently stands. Therefore, we invite you to submit a revised version of the manuscript that addresses the points raised during the review process.

Be sure to:

Align the main objective of study (SDG3 Strategies) with results and discussion section.Provide detailed practical insights and recommendation on SDG3 implementation in Italian region.Develop coherence from introduction to the conclusion and align each section.

We look forward to receiving your revised manuscript.

Kind regards,

Ali Junaid Khan, PhD

Academic Editor

PLOS ONE

Journal Requirements:

https://journals.plos.org/plosone/s/file?id=ba62/PLOSOne_formatting_sample_title_authors_affiliations.pdf ..

Reviewers' comments:

Reviewer's Responses to Questions

**Comments to the Author**

1. Is the manuscript technically sound, and do the data support the conclusions?

Reviewer #1: Yes

Reviewer #2: Yes

Reviewer #3: No

2. Has the statistical analysis been performed appropriately and rigorously? 

Reviewer #1: N/A

Reviewer #2: Yes

Reviewer #3: I Don't Know

3. Have the authors made all data underlying the findings in their manuscript fully available?

Reviewer #1: Yes

Reviewer #2: Yes

Reviewer #3: No

4. Is the manuscript presented in an intelligible fashion and written in standard English?

Reviewer #1: Yes

Reviewer #2: Yes

Reviewer #3: Yes

5. Review Comments to the Author

Reviewer #1: This manuscript is very well written and well organized. The method and result are clearly presented. Overall, this article scientificaly well written. The result clearly answer the research questions

Reviewer #2: I congratulate the authors. They have worked on an important topic. The research has significant potential for developing health policy, particularly in Italy. However, with some improvements, it could be transformed into a better structure.

Reviewer #3: 1. The abstract does not comprehensively reflect the full content of the manuscript and requires revision. It is recommended that the term "Sustainable Development Goals" be written out in full the first time it appears in the abstract, followed by its abbreviation (SDGs).

2. The introduction and background sections should be merged and summarized, in accordance with the journal’s formatting guidelines.

3. In the Data section, please specify the variables included in the ISTAT database, along with their characteristics and measurement scales.

4. Some of the content currently under the "Clustering Model" subsection in the Results should be moved to the Methods section, where it is more appropriate.

5. The rationale for selecting each indicator for each cluster, as well as the selection method, should be clearly explained in the Methods section.

6. The Discussion section is not well-developed. It primarily reiterates the results without offering sufficient interpretation, comparison with other studies, or practical implications for the key findings.

7. The Conclusion section is overly long. It should be shortened and focus on drawing conclusions directly based on the specific findings of this study.

8. The study’s limitations should be clearly stated, along with recommendations for future research.

9. A particularly important point is that the main aim of the study is to propose strategies for translating global and national policies and goals into localized interventions and actions, taking into account regional economic, social, and cultural differences. However, this core objective is scarcely addressed in the Results and Discussion sections and should be emphasized more clearly.

6. PLOS authors have the option to publish the peer review history of their article (what does this mean? ). If published, this will include your full peer review and any attached files.). If published, this will include your full peer review and any attached files.

**Do you want your identity to be public for this peer review?** For information about this choice, including consent withdrawal, please see our For information about this choice, including consent withdrawal, please see our Privacy Policy ..

Reviewer #1: No

Reviewer #2: **Yes:** Dr. Yaşar GÖKALPDr. Yaşar GÖKALP

Reviewer #3: No

While revising your submission, please upload your figure files to the Preflight Analysis and Conversion Engine (PACE) digital diagnostic tool, https://pacev2.apexcovantage.com/ . PACE helps ensure that figures meet PLOS requirements. To use PACE, you must first register as a user. Registration is free. Then, login and navigate to the UPLOAD tab, where you will find detailed instructions on how to use the tool. If you encounter any issues or have any questions when using PACE, please email PLOS at . PACE helps ensure that figures meet PLOS requirements. To use PACE, you must first register as a user. Registration is free. Then, login and navigate to the UPLOAD tab, where you will find detailed instructions on how to use the tool. If you encounter any issues or have any questions when using PACE, please email PLOS at figures@plos.org . Please note that Supporting Information files do not need this step.. Please note that Supporting Information files do not need this step.

---

## [Author Response · Author response to Decision Letter 1]

12 Sep 2025

All responses to the comments of Editor and reviewers are reported in the attached file "Response to reviewers".

---

## [Decision Letter · Decision Letter 1]

5 Oct 2025

PONE-D-25-29410R1Unveiling and Understanding Health Inequalities: A Bi-Clustering Study on SDG3 Implementation in the Italian regionsPLOS ONE

Dear Dr. Giancotti,

Thank you for submitting your manuscript to PLOS ONE. After careful consideration, we feel that it has merit but does not fully meet PLOS ONE’s publication criteria as it currently stands. Therefore, we invite you to submit a revised version of the manuscript that addresses the points raised during the review process.

Be sure to:

Please clearly address each comment of all reviewers. As one of reviewer is not satisfied with revised version and concerns of reviewer were not addressed.

We look forward to receiving your revised manuscript.

Kind regards,

Ali Junaid Khan, PhD

Academic Editor

PLOS ONE

Journal Requirements:

Reviewers' comments:

Reviewer's Responses to Questions

**Comments to the Author**

1. If the authors have adequately addressed your comments raised in a previous round of review and you feel that this manuscript is now acceptable for publication, you may indicate that here to bypass the “Comments to the Author” section, enter your conflict of interest statement in the “Confidential to Editor” section, and submit your "Accept" recommendation.

Reviewer #2: (No Response)

Reviewer #3: (No Response)

2. Is the manuscript technically sound, and do the data support the conclusions?

Reviewer #2: Partly

Reviewer #3: No

3. Has the statistical analysis been performed appropriately and rigorously? 

Reviewer #2: I Don't Know

Reviewer #3: I Don't Know

4. Have the authors made all data underlying the findings in their manuscript fully available?

Reviewer #2: Yes

Reviewer #3: No

5. Is the manuscript presented in an intelligible fashion and written in standard English?

Reviewer #2: Yes

Reviewer #3: Yes

6. Review Comments to the Author

Reviewer #2: (No Response)

Reviewer #3: I appreciate the authors’ efforts; however, the revisions requested in my previous review have not been addressed. The manuscript remains unchanged with respect to the major concerns raised, and the necessary corrections and clarifications have not been implemented.

7. PLOS authors have the option to publish the peer review history of their article (what does this mean? ). If published, this will include your full peer review and any attached files.). If published, this will include your full peer review and any attached files.

**Do you want your identity to be public for this peer review?** For information about this choice, including consent withdrawal, please see our For information about this choice, including consent withdrawal, please see our Privacy Policy ..

Reviewer #2: **Yes:** Yaşar GökalpYaşar Gökalp

Reviewer #3: No

While revising your submission, please upload your figure files to the Preflight Analysis and Conversion Engine (PACE) digital diagnostic tool, https://pacev2.apexcovantage.com/ . PACE helps ensure that figures meet PLOS requirements. To use PACE, you must first register as a user. Registration is free. Then, login and navigate to the UPLOAD tab, where you will find detailed instructions on how to use the tool. If you encounter any issues or have any questions when using PACE, please email PLOS at . PACE helps ensure that figures meet PLOS requirements. To use PACE, you must first register as a user. Registration is free. Then, login and navigate to the UPLOAD tab, where you will find detailed instructions on how to use the tool. If you encounter any issues or have any questions when using PACE, please email PLOS at figures@plos.org . Please note that Supporting Information files do not need this step.. Please note that Supporting Information files do not need this step.

---

## [Author Response · Author response to Decision Letter 2]

19 Oct 2025

Authors’ response to Reviewers’ comments (The responses to the reviewers’ comments are likewise provided in the attached file ‘Response to Reviewers’)

PONE-D-25-29410R1

Unveiling and Understanding Health Inequalities: A Bi-Clustering Study on SDG3 Implementation in the Italian regions

The authors of this paper would like to thank the reviewers of PLOS ONE for their insightful comments and constructive suggestions, which have helped us significantly improve the quality and clarity of our manuscript. Please note that the revised manuscript has been submitted in Track Changes mode (Revised Manuscript with Track Changes), allowing you to easily identify all modifications made in response to the reviewers’ feedback. Additionally, a clean version of the manuscript has been submitted under the filename “Manuscript”.

1. REVIEWER'S COMMENTS - METHODOLOGY:

Although details on normalization and missing data were added, the rationale for parameter choices (e.g., variance thresholds, similarity measures, algorithm settings) remains insufficient. These choices should be justified, including their potential effect on robustness.

AUTHORS' RESPONSES:

We thank the reviewer for the methodological suggestions.

Regarding parametrization of the algorithms and libraries used in the manuscript, we added in the paper the following part:

“…The analysis was performed using the Python Scikit-learn library, specifically the Spectracl Bi-clustering using the default settings, with the “scale” method for row/column normalization. This setting applies two different normalizations one over rows and another over columns, a choice justified by assuming that the variability internal to the indicators could be different of the variability along the regions. Finally, the K-Means methods have the only parameter n_clusters = 3. Before applying any clustering or bi-clustering algorithm data of each indicator have been transformed with z-score to have comparable scale for each variable. This to avoid the pre-dominance of some indicator given its unit of measure….”

(See section 2. Materials and Methods – 2.1 Biclustering technique)

2. REVIEWER'S COMMENTS - METHODOLOGY:

The most critical weakness is that cluster differences are presented only descriptively. To strengthen the validity of the results, they should be supported with statistical tests. For example:

Use ANOVA or t-tests to demonstrate whether clusters differ significantly on SDG3 indicators.

Apply post-hoc tests (e.g., Tukey’s HSD) to identify which clusters differ.

Report effect sizes and p-values to establish the meaningfulness of the clusters.

Without such validation, it is unclear whether the clusters are statistically distinct or artifacts of the algorithm

AUTHORS' RESPONSES:

We approved this comment and integrated the paper as follows:

“….We used the ANOVA f-test to assess for significant differences between cluster means. The results of these tests, detailed in the Supporting Information (Table S2, S3), provide a clear comparison:

- K-Means divides regions based on all health indicators, creating groups with similar means and standard deviations. In fact, the algorithm works by minimizing the within-cluster variance (inertia) by finding the centroids of data points using Euclidean distances [59].

- Bi-clustering, instead, by allowing for partial expression of variables, is able to separate not only Italian regions (rows), but also SDG3 health indicators (columns) making it feasible to create more independent clusters where independence can be seen (at least partially) with ANOVA tests and the Tukey-HSD group test.

In summary, statistical tests show that Bi-clustering builds better separation, and (partial) cluster independence (see for more details Table S4- S7 in the Supporting Information, where cluster 0 is different from cluster 1 and 2, but cluster 1 is not independent from cluster 2, with ANOVA and Tukey-HSD). Conversely, K-Means tends to rearrange members in order to have a smaller within - cluster variance (inertia) and a general smaller in-cluster variance renouncing to have independent clusters.

…..

(See section 2. Materials and Methods - 2.1 Bi-clustering technique).

Morevover, we performed the post-hoc Tukey HSD test (S6-S7) following the ANOVA test (S4-S5) for independent samples to each bi-cluster and to each cluster obtained with kMeans. We used these tests to study the independence of the clusters in both cases. The Null hypothesis for Anova test being that the means of all the 3 groups are equal (i.e., there's no difference between the groups).

While for Tukey HSD the null hypothesis being that the means of the two groups being compared are equal i.e. Tukey works by comparing pairs and not the full groups.

While the full statistics are shown in the tables S4 to S7 (see Supporting Information), for the convenience of the reader, the summary of the p-values of the Tukey HSD, for bi-clustering and for kMeans is shown in the Table below (see table S6-S7 in the Supporting Information for the full version):

Clusters BiClustering Kmeans

(0, 1) 0.024** 0.929

(0, 2) 0.001*** 0.999

(1, 2) 0.222 0.975

Each pair of clusters 0,1 or 0,2 has the corresponding p-value for the test statistics in the two cases BiClustering and kMeans. When the p-value is lower than 0.05 we can conclude that the two clusters have a significant difference in the individual means.

In synthesis for the bi-clustering we obtain significant differences in the combinations 0,1, 0,2, 1,0, 2,0 but not in the case 1,2 and 2,1 clusters. Both tests (ANOVA and Tukey-HSD) confirm the results with p-values that are lower than 0.05 except for the 1,2 pair of clusters.

Instead, in the case of K-Means for both tests (ANOVA and Tukey-HSD) there is no independent set of clusters as the p-value is always (much) larger than 0.05.

There is likely much more relevance and information in the bi-clusters rather than in the k-Means clustering, a confirmation that the partial expression is able to recover non obvious patterns in the dataset.

We conclude that one of the strengths of the bi-clustering is its ability to get patterns that, being expressed partially, could be less noisy and more clear to reveal.

(See Supporting Information)

3. REVIEWERS' COMMENTS - MOTIVATION:

The rationale for using bi-clustering remains weak. Please clarify why this method was chosen over conventional techniques (k-means, hierarchical clustering, PCA) and what unique insights it offers in the SDG3 context.

AUTHORS' RESPONSES:

Thanks for the suggestion. We clarify this point, integrating the paper as follows:

“…..bi-clustering is not just a novel application; it is a promising methodological tool that generates more robust and statistically reliable findings because it overcomes limitations inherent in both traditional econometric methods (like panel analysis) and standard unsupervised clustering (like k-Means), making it a promising tool for regional-level research. Specifically, compared to standard panel data methods, bi-clustering offers major robustness by avoiding restrictive assumptions relative to:

• Assumption-Free Analysis: Bi-clustering requires no assumptions regarding linearity, independence, or inner correlation of the data. This eliminates the necessity for complex diagnostic testing (such as those for collinearity or endogeneity) that often complicates and biases estimations in econometric panel models.

• Time-Insensitivity: It is highly effective in contexts where the time dimension is insufficient for reliable panel analysis, or where the research focus is primarily on spatial heterogeneity rather than temporal dynamics.

Moreover, bi-clustering significantly enhances pattern discovery by leveraging the concept of “partial expression,” which is crucial when variables are numerous, and observations are relatively scarce (both spatially and temporally) as follows:

• Bidimensionality for Less Noise: Traditional methods (like k-Means) create unidimensional clusters (grouping regions based on all indicators), forcing relationships to hold across the full dataset. In contrast, Spectral Bi-clustering [55] simultaneously clusters rows (Italian regions) and columns (SDG3 indicators). This bi-dimensionality ensures that bi-clusters only include SDG3 indicators that are more similar and constant (low variance) within that specific subset of Italian regions. This focused approach systematically reduces noise and maximizes pattern coherence.

• Capturing Localized Phenomena: This ability to discover “partial expressions” is key for regional research. A variable (e.g., an SDG3 health indicator) may be highly relevant in a subset of Italian regions, but not across the entire Italy. Bi-clustering inherently captures these localized relationships (e.g., specific regional dependencies, like border regions benefiting from foreign health facilities) that would otherwise be masked by national averages or require the manual addition of complex dummy variables in traditional models…..”

(See section 2. Materials and Methods - 2.1 Bi-clustering technique).

4. REVIEWER'S COMMENTS - LITERATURE REVIEW:

The literature review has been improved but still does not clearly define the research gap. Please explicitly state what prior studies have done, where they fall short, and how your study differs.

AUTHORS' RESPONSES:

According to this comment, the paper was integrated as follows:

“….Evidence on SDG implementation points to difficulties in coordination and integration and highlights the absence of operational guidance for translating global goals into subnational action plans: goals are often viewed as too broad, focusing on national and global averages without accounting for significant within-country distributional inequalities [14,15,16].….

…. much of the empirical assessment of SDG 3 progress has concentrated on national scores or cross-country comparisons which—though informative—mask within-country heterogeneity crucial for designing appropriate policies [11,12,17]. Current SDG 3 monitoring efforts largely aggregate indicators to country means or produce dashboards that do not systematically identify “where” and “which” targets are underperforming within countries [4,10,17].

In practical terms, policymakers need evidence and methods to (i) identify groups of territories that share similar challenges in achieving SDG 3 targets, and (ii) isolate the indicators most responsible for underperformance within each group. Prior EU-wide and national case studies either (a) remain at the national scale, or (b) describe subnational disparities without pinpointing the subset of SDG 3 indicators that most strongly differentiate such groups [17,19]. This leaves a clear research gap: subnational, indicator-sensitive analyses are needed to go beyond averages and to reveal “which indicators matter, where” in a way that is directly actionable for regional planning. This need is particularly acute in countries with decentralized health systems and marked regional differences [22,23,24,25,26,27,28]. ….

…. it is essential that research examine differences among regional health systems to understand how global objectives such as SDG 3 can be effectively translated into meaningful social change. By monitoring health outcomes at the regional level, policymakers can identify obstacles to SDG implementation and promote effective, equity-oriented policies. In this sense, there is a gap in the literature that this study seeks to fill by offering a subnational perspective that responds to calls to move beyond averages and to identify context-specific barriers and enablers for the implementation of SDG 3 in Italy.”

(See section 1: Introduction)

5. REVIEWER'S COMMENTS - LITERATURE REVIEW:

Include more recent SSCI/SCI-indexed works and international reports, especially those discussing design–outcome relationships, financing barriers, and local variation.

AUTHORS' RESPONSES:

Based on this feedback, we made the following updates:

“….Recent Italian and WHO reports—documenting the persistence of territorial gaps in health outcomes—confirm the relevance of regional-scale analyses to support policy development and decision-making appropriate to diverse territorial contexts [31,46]. The recent ISTAT report likewise underscores the centrality of a subnational reading of the SDGs in Italy, particularly Goal 3 [47]. In line with the “no one left behind” principle, it reinforces the focus on territorial inequalities. Regional analysis shows a heterogeneous picture, with generally better performance in the North-East and delays in the South and Islands; despite signs of recent improvement in some southern regions, progress still does not bridge the historical gaps with the North. These findings strengthen the case, in the Italian context, for regional analyses and targeted policies to fully translate SDG 3 into equitable and context-sensitive health outcomes [47].

In addition, recent Italian evidence has reinforced the link between the “design” of health services and health outcomes, while highlighting pronounced local variation [48,49]. Studies also point to a non-negligible organizational–geographic component, suggesting that differences in the design and management of regional services (e.g., supply, waiting times) translate into disparities in utilization and, ultimately, outcomes. Finally, evidence on fiscal and organizational decentralization suggests a risk of amplifying pre-existing inequalities in access to, availability and use of services, and in health outcomes, when resources are devolved without a full understanding of regional characteristics, needs, and gaps [50].

This recent evidence—centered on design–outcome relationships, financial barriers, and local variation—justifies our subnational approach……”

(See section 1: Introduction)

These are the integrated references:

- Sarti, F. M. (2023). Challenges in assessment of health systems decentralization: the role of path dependence and choice of indicators. International Journal of Health Policy and Management, 12, 01-10. [citation 50]

- Naghavi, M., Zamagni, G., Abbafati, C., Armocida, B., Agodi, A., Alicandro, G., … & Monasta, L. (2025). State of health and inequalities among Italian regions from 2000 to 2021: a systematic analysis based on the Global Burden of Disease Study 2021. The Lancet Public Health, 10(4), e309-e320. [citation 49]

- Matranga, D., & Maniscalco, L. (2022). Inequality in healthcare utilization in Italy: how important are barriers to access? International Journal of Environmental Research and Public Health, 19(3), 1697. [citation 48]

- World Health Organization. (2024). Aligning for country impact: 2024 progress report on the Global Action Plan for Healthy Lives and Well-being for All. World Health Organization. [citation 46]

- ISTAT REPORT 2024, available at https://www.istat.it/wp-content/uploads/2024/12/2024-SDGs-Report-Ebook.pdf [citation 47]

To the best of our knowledge, there are no recent studies sufficiently aligned with the objectives and conceptual framing of our research to warrant integration into the manuscript.

6. REVIEWER'S COMMENTS - RESEARCH QUESTION:

The research question is implied but not explicitly stated. Please formulate it clearly (e.g., “How can bi-clustering of SDG3 indicators reveal distinct groups of countries, and what policy implications follow?”).

AUTHORS' RESPONSES:

In response to this comment, the manuscript has been revised as follows:

“….Specifically, we apply an innovative data-mining technique—bi-clustering [51]—to ISTAT’s regional SDG 3 indicators (2013–2019) in order to: (1) identify groups of Italian regions with similar SDG 3 profiles; (2) determine, for each group, which indicators are most salient (i.e., most strongly expressed) and how they deviate from national averages; and (3) translate these findings into cluster-tailored implications for health policy. These aims are articulated in two research questions (RQs):

1. How can bi-clustering of SDG 3 indicators reveal distinct groups of Italian regions, and what insights do these clusters provide about regional health-related trends? The innovation of the bi-clustering algorithm lies in its ability

---

## [Decision Letter · Decision Letter 2]

17 Nov 2025

PONE-D-25-29410R2Unveiling and Understanding Health Inequalities: A Bi-Clustering Study on SDG3 Implementation in the Italian regionsPLOS ONE

Dear Dr. Giancotti,

Thank you for submitting your manuscript to PLOS ONE. After careful consideration, we feel that it has merit but does not fully meet PLOS ONE’s publication criteria as it currently stands. Therefore, we invite you to submit a revised version of the manuscript that addresses the points raised during the review process.

We look forward to receiving your revised manuscript.

Kind regards,

Ali Junaid Khan, PhD

Academic Editor

PLOS ONE

Journal Requirements:

Reviewers' comments:

Reviewer's Responses to Questions

**Comments to the Author**

1. If the authors have adequately addressed your comments raised in a previous round of review and you feel that this manuscript is now acceptable for publication, you may indicate that here to bypass the “Comments to the Author” section, enter your conflict of interest statement in the “Confidential to Editor” section, and submit your "Accept" recommendation.

Reviewer #2: All comments have been addressed

2. Is the manuscript technically sound, and do the data support the conclusions?

Reviewer #2: Yes

3. Has the statistical analysis been performed appropriately and rigorously? 

Reviewer #2: Yes

4. Have the authors made all data underlying the findings in their manuscript fully available?

Reviewer #2: Yes

5. Is the manuscript presented in an intelligible fashion and written in standard English?

Reviewer #2: Yes

6. Review Comments to the Author

Reviewer #2: Reporting of ANOVA results:

The methodology section states that “post-hoc tests were performed (one-way ANOVA, p < 0.05)”; however, the results are not shown in the tables. It is recommended that the authors add basic statistics such as the F-value, p-value, and effect size (η²) for each cluster to the table or present them as supplementary material. This would allow readers to verify the statistical validity of the cluster differences.

7. PLOS authors have the option to publish the peer review history of their article (what does this mean? ). If published, this will include your full peer review and any attached files.). If published, this will include your full peer review and any attached files.

**Do you want your identity to be public for this peer review?** For information about this choice, including consent withdrawal, please see our For information about this choice, including consent withdrawal, please see our Privacy Policy ..

Reviewer #2: **Yes:** Yaşar GÖKALPYaşar GÖKALP

---

## [Author Response · Author response to Decision Letter 3]

23 Nov 2025

Reviewer #2’s comment

Methodology - Reporting of ANOVA results.

The methodology section states that “post-hoc tests were performed (one-way ANOVA, p < 0.05)”; however, the results are not shown in the tables. It is recommended that the authors add basic statistics such as the F-value, p-value, and effect size (η²) for each cluster to the table or present them as supplementary material. This would allow readers to verify the statistical validity of the cluster differences.

Authors’ responses.

We thank the reviewer for the methodological suggestions.

Basic statistical values are detailed in the Supporting Information: the F-value and p-value are presented in the Table S4 and S5, while the effect size is reported in the Table S8.

The text was integrated as follows:

“…We used the ANOVA f-test to assess for significant differences between cluster means (Table S4, S5 in the Supporting Information show F-value and p-value). In addition, we performed, for the same goal, the Kruskall-Wallis test to provide an alternative comparison (Table S2, S3 in the Supporting Materials)…

…. Moreover, the K-Means methods have the only parameter n_clusters = 3. We remember that before applying any clustering or bi-clustering algorithm data of each indicator have been transformed with z-score to have comparable scale for each variable. This to avoid the pre-dominance of some indicator given its unit of measure. Finally, we investigated the “size effect” of each cluster by computing the Cohen’s d effect size measure for bi-clustering and k-means. Based on this metric, the bi-clustering results yield stronger distinctions: all bi-clusters demonstrated at least a medium effect size, with clusters 0 and 2 achieving a larger effect size. Conversely, all k-Means clusters show a small effect size, indicating that their group averages are too similar to indicate a meaningful difference (Table S8 in the Supporting Materials)..”

(See section 2.1 Bi-clustering technique)

---

## [Decision Letter · Decision Letter 3]

21 Dec 2025

Unveiling and Understanding Health Inequalities: A Bi-Clustering Study on SDG3 Implementation in the Italian regions

PONE-D-25-29410R3

Dear Dr. Giancotti,

We’re pleased to inform you that your manuscript has been judged scientifically suitable for publication and will be formally accepted for publication once it meets all outstanding technical requirements.

Kind regards,

Ali Junaid Khan, PhD

Academic Editor

PLOS One

Additional Editor Comments (optional):

Reviewers' comments:

Reviewer's Responses to Questions

**Comments to the Author**

1. If the authors have adequately addressed your comments raised in a previous round of review and you feel that this manuscript is now acceptable for publication, you may indicate that here to bypass the “Comments to the Author” section, enter your conflict of interest statement in the “Confidential to Editor” section, and submit your "Accept" recommendation.

Reviewer #2: All comments have been addressed

2. Is the manuscript technically sound, and do the data support the conclusions?

Reviewer #2: Partly

3. Has the statistical analysis been performed appropriately and rigorously? 

Reviewer #2: N/A

4. Have the authors made all data underlying the findings in their manuscript fully available?

Reviewer #2: Yes

5. Is the manuscript presented in an intelligible fashion and written in standard English?

Reviewer #2: Yes

6. Review Comments to the Author

Reviewer #2: In my previous report, the issues raised regarding the abstract, originality, methodology, literature review, and methods sections have not been fully addressed. The revisions requested in these areas remain insufficient. My previous requests therefore still stand

7. PLOS authors have the option to publish the peer review history of their article (what does this mean? ). If published, this will include your full peer review and any attached files.). If published, this will include your full peer review and any attached files.

**Do you want your identity to be public for this peer review?** For information about this choice, including consent withdrawal, please see our For information about this choice, including consent withdrawal, please see our Privacy Policy ..

Reviewer #2: **Yes:** Yaşar GÖKALPYaşar GÖKALP

---

## [Editor Report · Acceptance letter]

PONE-D-25-29410R3

PLOS One

Dear Dr. Giancotti,

I'm pleased to inform you that your manuscript has been deemed suitable for publication in PLOS One. Congratulations! Your manuscript is now being handed over to our production team.

Kind regards,

on behalf of

Dr Ali Junaid Khan

Academic Editor

PLOS One